# Preparation, Properties and Mechanisms of Carbon Fiber/Polymer Composites for Thermal Management Applications

**DOI:** 10.3390/polym13010169

**Published:** 2021-01-05

**Authors:** Zulfiqar Ali, Yuan Gao, Bo Tang, Xinfeng Wu, Ying Wang, Maohua Li, Xiao Hou, Linhong Li, Nan Jiang, Jinhong Yu

**Affiliations:** 1Key Laboratory of Marine Materials and Related Technologies, Zhejiang Key Laboratory of Marine Materials and Protective Technologies, Ningbo Institute of Materials Technology and Engineering, Chinese Academy of Sciences, Ningbo 315201, China; zali5529@gmail.com (Z.A.); limaohua@nimte.ac.cn (M.L.); houxiao@nimte.ac.cn (X.H.); lilinhong@nimte.ac.cn (L.L.); 2Center of Materials Science and Optoelectronics Engineering, University of Chinese Academy of Sciences, Beijing 100049, China; 3College of Ocean Science and Engineering, Shanghai Maritime University, Shanghai 201306, China; 201830410075@stu.shmtu.edu.cn (Y.G.); luoyahangzhou@163.com (B.T.); 201940110009@stu.shmtu.edu.cn (Y.W.)

**Keywords:** carbon fiber, polymer composite, thermal conductivity

## Abstract

With the increasing integration and miniaturization of electronic devices, heat dissipation has become a major challenge. The traditional printed polymer circuit board can no longer meet the heat dissipation demands of microelectronic equipment. If the heat cannot be removed quickly and effectively, the efficiency of the devices will be decreased and their lifetime will be shortened. In addition, the development of the aerospace, automobiles, light emitting diode (LED{ TA \1 “LED; lightemitting diode” \s “LED” \c 1 }) and energy harvesting and conversion has gradually increased the demand for low-density and high thermal conductive materials. In recent years, carbon fiber (CF{ TA \1 “CF; carbon fiber” \c 1 }) has been widely used for the preparation of polymer composites due to its good mechanical property and ultra-high thermal conductivity. CF materials easily form thermal conduction paths through polymer composites to improve the thermal conductivity. This paper describes the research progress, thermal conductivity mechanisms, preparation methods, factors influencing thermal conductivity and provides relevant suggestions for the development of CF composites for thermal management.

## 1. Introduction

With the advancement of technology, innovative electronic equipment is emerging rapidly. The idea of multi-functionality of a device in a smaller footprint has become a trend which causes device heat-up and affects the equipment operation, efficiency, steadiness and life-time. Thus, heat dissipation through electronic devices is a counterpart of technological advancement. Polymers are the most preferred materials for use in electronics due to their easier fabrication and availability, lower cost, compactness and lighter weight. However, the low thermal conductivity of polymers due to their amorphous structure limits their usage for thermal management systems. Thus, the inclusion of high thermal conductivity fillers in polymers to increase their thermal conductivity by forming polymer composites has become a hot topic of research in the recent era [1,2,3,4,5,6]. Figure 1 shows schematic diagram of factors, which may include CF type, content, reinforced thermal fillers, surface modification, orientation and CF 3D pathways, having significant impact on the thermal conductivity of the CF composites.

Carbon fiber (CF) was first synthesized by the English scientist Joseph Swan in the 18th century (1860). Later, in 1879 the American scientist Thomas Edison showed a high temperature cotton/bamboo CF produced by a carbonization method. These attempts were made for manufacturing carbon fiber-based filaments for glowing light bulbs by heating the CF filament with electricity. Carbon fibers capable of lower stiffness and lower strength with only twenty percent (20 wt%) carbon content were reinvented in 1958. To impart efficient properties composites with high carbon content were made by using innovative manufacturing techniques. Thus, by usage of the organic polymer polyacrylonitrile (PAN{ TA \1 “PAN; polyacrylonitrile” \s “PAN” \c 1 }) a carbon fiber with 55 wt% carbon content was created by Shindo. In 1960 carbon fibers containing 99 wt% carbon content were developed by utilizing a cellulose fiber. Due to their high concentration of carbon, these carbon fiber were capable of higher strength to weight ratio and higher flexibility with greater potential to be used for composite preparation for great electronic applications, in contrast to previously manufactured CFs and other materials [11,12,13,14,15].

Typically carbon fiber is defined as a fiber composed by long chains of carbon atoms, which consists of about 90 percent or more carbon such that the C-atoms are bonded with each other and have diameters in the range of 5 to 10 micrometers. Carbon fiber, also known as graphite fiber, is preferred in heavy industries, e.g., aerospace engineering, sports, automobiles, etc. due to its significant properties related to size, high strength, flexibility, stiffness and temperature [11,13,16,17]. A composite is defined as a mishmash of two different materials having different physical and chemical properties to obtain superior properties as compared to the individual constituent material properties, while the elements individually do not lose their uniqueness in this amalgamation. Usually these two different materials are known as the fiber and the matrix and the resultant composite is known as a fiber composite matrix. Fiber provides strength and stiffness while the matrix shelters it from environmental damage [18,19,20,21,22]. With technology advancement, high performance devices are emerging rapidly and the heat dissipation problem directly affects the lifetime and performance of these devices. Thus there is an urgent need for composites helpful for heat removal in devices. As carbon fibers are rich in carbon content, this indirectly affects the thermal conductive performance of carbon fibers depending upon their orientation. For fulfilling the requirement of thermal conductive material to be used in high performing devices, carbon fiber is used in many matrices as a filler. CF-based copper composites hybridized by impregnation of epoxy were synthesized and cured by rapid thermal annealing, and as a result a higher thermal conductivity was noted which was eighteen times higher in the in parallel direction and six times in the perpendicular direction, respectively, however the sample was oriented by a compact molding technique [23,24,25].

Compositing an ultra-high thermally conductive CF with a polymer matrix and reinforced thermal conductive fillers has become the mainstream direction of scientific researchers at this stage. The thermal conductivity of the polymer matrix can be significantly improved by maintaining good mechanical properties. This review summarizes different studies exploring CF composites’ properties and their preparation methods, as well as the factors influencing the thermal conductivity of CF fiber composites. Previous studies on thermally conductive composites usually involved adding oxides (Al_2_O_3_, MgO), nitrides (Si_3_N_4_, BN, AlN) or carbide (SiC) particles to the polymer matrix. To some extent, the thermal conductivity of composite is thus improved, but it leads to destruction of the mechanical properties [26,27,28]. CF material has become a good choice in the field of thermal management. The density of CF (1.5~2.0 g/cm^3^) is lower than that of aluminum, but it has excellent mechanical properties and thermal conductivity, and it is easy for CF to form thermal conduction paths inside the composite material. At present, combining with CF and polymer material has become the mainstream in the field of preparing thermally conductive composites.

Wei et al. [29] mixed pitch-based CF and polydimethylsiloxane (PDMS) thoroughly to prepare a CF/PDMS thermal conductive composite. When the CF content reached 20 wt%, its thermal conductivity (2.73 W/m·K) was about 17.1 times that of pure PDMS and the bending strength and compressibility of the composite material were not affected. When the CF forms a thermal conduction path in the polymer matrix, it promotes the conduction of heat inside the material, which is beneficial to improve the thermal conductivity. However, relying only on a simple mixing method to synthesize CF/polymer composites to form a systematic guiding structure is challenging, which limits the possible thermal conductivity improvement [30]. For this purpose, many researchers have used electrostatic flocking or freeze-drying methods to make CFs present a guiding structure distribution in the polymer, further improving the internal thermal conduction path. Ma et al. [7] put the CF suspension under a low temperature (−48 °C) and low pressure (27 Pa) environment for 50 h by a freeze-drying method to prepare a three dimensional (3D) CF skeleton structures with different contents. When the CF content was 13 wt%, the vertical thermal conductivity of the composite was about 15 times that of the pure epoxy (2.84 W/m·K). Moreover, Hou et al. [8] proposed a CF with a guiding structure with PDMS made using a vacuum impregnation method. When the CF content was 12.8 vol%, the thermal conductivity of the composite reached 6.04 W/m·K in the vertical direction. The reed structure CF formed a complete heat conduction chain which could effectively improve the heat transfer of the composite material in the direction of the CF. However, it had a small impact on the thermal conductivity of the composite material in the plane direction, which also limited the CF composite’s application. In order to solve this problem, a low content of reinforced fillers are added into the matrix that acted as a bridge to connect the CF. Li et al. [31] dipped CF bundles in an acetone solution containing an epoxy resin. The impregnated CF tow was firmly coated on a steel frame structure under the action of tension. After curing at 120 °C, it was composited with a graphite sheet by lamination to produce a 3D CF thermal conductive composite. As the number of graphite sheets increased, the thermal conductivity of the CF composite material could reach 6.2 W/m·K in the vertical direction. Wang et al. [32] added Al_2_O_3_ particles to CF/epoxy resin and mixed them well. When the mass fractions of CF and Al_2_O_3_ were 6.4 wt% and 74 wt%, respectively, the thermal conductivity of the composite was 3.84 W/m·K, which was about 20 times that of the pure epoxy. The addition of the reinforced fillers makes the thermal conductive chains form a thermally conductive network. This 3D thermal structure promotes the improvement of the thermal conductivity of the CF composite material. However, the amount of thermal conductive fillers and the binding of polymer matrix and CF are still the research focus on CF thermal management. The adjustment of carbon fiber into the matrices is usually into two different forms, named as continuous and discontinuous [33]. This literature review studies the thermal conductivity of polymer composites and more specifically, the effects and importance of carbon fiber as a thermally conductive filler. Different thermal polymer composites containing carbon fiber filler were compared and the thermal conductivity results were analyzed. Different preparation methods are implemented by different researchers and also same materials may differ regarding shapes, structures and size thus comparisons are difficult but the purpose of this review is to introduce thermal conductivity mechanisms, different preparation methods and their impact on thermal conductivity results and some influencing factors of thermal conductivity of CF/polymer composites are also discussed. The schematic diagram below illustrates the advantages of carbon fiber as a filler which include high orientation capability of CF through polymers, continuity, the ability to show high thermal conductivity by synergistic effects with other fillers present in polymers, the ability to show property modification phenomena and the provision of 3D conductive pathways even at low loading.

## 2. Thermal Transport Mechanisms

Usually heat is circulated in three different ways—conduction, convection and radiation—but conduction is regarded as a more satisfactory way for heat dispersion. Recalling the traditional mechanism of heat conduction in matter, it can be illustrated as when heat is introduced to one atom in matter it starts vibrating and collisions takes place between the atoms in the matter and gradually heat is delivered through whole substance [34].

As thermal conductivity is a property of material that is directly related to its composition, crystalline materials have the highest thermal conductivity as they are composed by ordered patterns of atoms, which provide a smooth pathway for heat dispersion. Heat is transported by a wave mechanism in crystalline materials, i.e., when heat influences a crystalline material surface then the atoms acquire vibrational energy and propagate this energy towards the adjacent atoms and thus heat is transported in a regular way to the whole crystalline material and finally heat is emitted to the environment. This is how the thermal conductivity mechanism works in crystalline materials [35,36,37]. Amorphous polymers are different compared to crystalline polymers, as they are composed of disconnected atoms having no specific patterns or arrangement thus the conduction of heat through such material varies in different situations. Due to the disordered atoms the heat is not dispersed resembling a wave, as the vibrational energy of atoms is also disrupted thus the speed of heat spread is also affected [37,38,39,40]. The mechanism of thermal conductivity in CF composites is shown in Figure 2.

Research on high thermal conductivity polymer composites is becoming a hot topic. Many researchers have focused and analyzed the thermal flow and its affecting factors in different composites and thus it was concluded that heat is transmitted in form of packets thus phonons are a central part of any discussion about the thermal conductivity mechanisms of composites. As discussed, crystalline materials possess high thermal conductivity values due to lattice atomic vibrations but these vibrations are not regular in all crystalline materials, as the regularity of atomic vibrations is dependent on various factors, e.g., discontinuities, defects, dislocations in a crystalline material are problematic for atomic vibrations and thus result in a reduction of the thermal conductivity. These imperfections in the crystallinity give birth to thermal resistance which causes the interruption of phonon transmission. There are several types of defects discussed in different literatures, which may classified as point, line and planar defects. There are some more types e.g., sheet defects and volume defects, as well [41,42,43,44,45,46].

Carbon nanotube and carbon fiber-based three-dimensional hybrid composites reinforced by silicon carbide (SiC) have been synthesized by Feng et al. in such a way that vertically aligned carbon nanotubes (VACNT) were developed on a carbon fiber (CF) surface. It was observed that the thermal conductivity was improved in comparison to carbon fiber-based SiC composites. The improvement was due to the inclusion of VACNTs which provided additional pathways for phonon dispersion [46]. The thermal conductivity enhancement is greatly dependent upon the aspect ratio of fillers in the matrices. Epoxy resin is a continuous but slow source of heat conduction which is boosted by the inclusion of fillers and it was revealed that larger size fillers are more effective in contrast to smaller size fillers for thermal conductivity enhancement, i.e., micro-fillers offer better results than nanofillers. According to this hypothesis the particle-particle distance matters in the heat conductivity boost and it can be lowered by using a high filler loading. Figure 2 shows different cases for thermal conductivity enrichment. It is clearly shown that long micro-fillers with high aspect ratio provide dramatic thermal conductivity improvement results [40].

## 3. Preparation Methods

### 3.1. Blending Method

The blending method is a very simple and low-cost method where the preparation process differs according to the CF lengths. When the CF length is short, the preparation steps of the CF composites are as follows: First, the reinforced thermal fillers are added to the polymer matrix, and a certain amount of short CF material is added to the mixture after magnetic stirring. Adding a small amount of modifier to the mixture is of great significance to reduce the viscosity of the polymer matrix which helps to reduce the air bubbles formed during the stirring process to avoid material defects. A schematic diagram is shown in Figure 3.

The improvement of the bonding performance between the CF and the polymer matrix helps to reduce the internal thermal resistance. When the length of the CF is long, the CF filaments overlap each other and form a 3D oriented structure. The CF felt with a 3D porous structure has a certain guide structure in the plane direction and is closely combined in the vertical direction to make the felt material have a certain integrity. The CF felt made of long CF filaments has been a great choice to make CF thermally conductive composites at this stage. Xu et al. [48] immersed acid-oxidized CF felt in a graphene oxide solution, and the CF felt with graphene oxide on the surface was reduced and composited with PAI resin. When the content of graphene and CF is 4.25 wt%, the thermal conductivity of the composite material is 0.53 W/m·K. From the SEM images, it can be seen that the graphene nanosheets adhered to the surface of the CF filaments and formed a thermal conduction path between the CF filaments. When the CF felt structure is relatively loose, covering the felt body with a copper cloth helps protect the felt guide structure from being damaged, which is beneficial to maintain the integrity of the thermal conductive model. When the carbon fiber is a felt material, unlike short carbon fibers, the suction filtration method is also an effective choice for the preparation [49,50]. Figure 3 shows the schematic process for combination of a carbon precursor polymer (CPP) and a thermally decomposable polymer [47].

### 3.2. Freeze-Drying Method

The CFs prepared by freeze-drying are distributed in an orderly way in the polymer matrix and form a certain guiding structure, which is beneficial to the formation of heat conduction networks inside the material [51]. Because of the simple operation method and low consumption cost, the freeze-drying method is not only the main method in industrial production, but also widely used in the laboratory [7]. The steps of the freeze-drying method are as follows [52]: First, the CF is blended in an aqueous solution. In this process, an appropriate amount of binder can be added to enhance the correlation between the CFs. the fully stirred carbon fiber suspension is poured on a copper module and kept in a container filled with liquid nitrogen such that liquid nitrogen should not be directly in contact with sample thus the orientation of CF in the mixture is guided by controlling the growth direction of ice crystals. Hou, et al. [8] synthesized carbon fiber-based PDMS composites for acquiring thermal conductivity boosted values by using a freeze drying method, as shown in Figure 4.

In view of the excellent characteristics such as simple operation, low cost and strong orientation of the carbon fiber skeleton, the freeze-drying method is widely used in the manufacture of carbon fiber thermal conductive composites. How to enhance the correlation between carbon fibers is also the key to further optimizing the carbon fiber skeleton structure [53]. Carbon fiber-MXene foam having vertically aligned carbon fibers was used to synthesize three dimensional (3D) CF-MXene/epoxy composites by using 30.2 wt% hybrid filler with freeze drying method and it was demonstrated that thermal conductivity (9.68 W/m·K) was enhanced 4509% in comparison to neat epoxy [9]. Carbon fiber and silica gel-based consolidated composites were prepared by using a freeze-drying method for obtaining higher thermal conductivity. Results showed that 13.4 times higher thermal conductivity (1.66 W/m·K) was obtained in contrast to pure silica gel. It was observed that with increasing content of carbon fiber (CF), the thermal conductivity values also increased [54].

### 3.3. Electrostatic Flocking Method

Electrostatic flocking is another simple and cheap method for CF/polymer composite preparation. CFs can easily form a guiding model with 3D structures in an electric field. The main process of electrostatic flocking is as follows [55]: The carbon fiber is in contact with the negative plate during the landing, and the negatively charged carbon fiber drops under the attraction of the positive plate. The CF is inserted vertically into the adhesive, coated on the positive plate under the action of the electric field [56]. Sun et al. adopted an electrostatic flocking method and developed a fuzzy mat based on installation of different fillers on a mat of carbon fiber and as a result high thermal conductivity enhancement was observed in the out-of-plane direction [57]. The schematic process is shown in Figure 5.

The electrostatic flocking method was mainly used in the preparation of embroidery and adsorption filter materials as a new production process. Nowadays, the electrostatic flocking method is not widely used in the field of CF thermal conductive composites, but this method is simple and can produce a 3D structured carbon fiber skeleton, which creates great conditions for the heat transfer [57]. There are many articles which highlight the improved arrangement of fibers and properties, improved by using electrostatic flocking method [58,59,60].

### 3.4. Vacuum Impregnation Method

In this method, the polymer matrix and CF are added after mixing the thermally conductive fillers with a solven and the mixture is impregnated under vacuum [61]. The vacuum impregnation method will promote the immersion of the polymer matrix into the porous CF network. Cf/ZrC-SiC composites were prepared by a vacuum impregnation technique and the adopted schematic process was as follows: the carbon fiber was modified by coating pyrolytic carbon (PyC) firstly, then 3D CF was impregnated by ZrC-SiC under vacuum. Later, a compacting process was done by applying 30 MPa pressure for two hours, and the composite was subjected to a drying process under vacuum at 80 °C and finally the sample was spark plasma sintered at 1800 °C [62]. Carbon fiber-based phase change composites (PCCs) were synthesized using a method in which the carbon fiber were orientated vertically by using a vacuum impregnation method, It was demonstrated that the carbon fiber acted as a supporting scaffold for anisotropic thermal conductivity enhancement and shape stability. The schematic process of vacuum impregnation of CF-based PCCs is shown in Figure 6.

Yao et al. [64] pre-sprayed the surface of a CF bundle with polycarbonate resin (PC). Experiments indicated that the surface bonding performance of the CF composite material pre-sprayed with PC was improved. When the thickness of the pre-sprayed PC did not reach 0.15 μm, the impregnation efficiency of CF is low and the resin cannot be completely impregnated into the CF. However, as the thickness increased, the interface bonding performance between the PC and the CF bundles also increased. Adding a certain amount of the modifier to the polymer matrix could reduce the viscosity of the polymer matrix to improve the impregnation efficiency, which is favorable to improve the mechanical properties and thermal conductivity of the composite material. Jun et al. [65] selected polyethylene glycol methyl methacrylate (PEGMEMA) and benzyl methacrylate (BZMA) as modifier monomers to add to the diglycidyl ether of bisphenol-A (DGEBA) resin. When the contents of PEGMEMA and BZMA modifiers were 16.7 and 3.3 wt%, respectively, the fracture toughness of the CF composite material reached its highest value, which indicates that with the addition of the modifier, the binding performance of CF and polymer matrix is enhanced. The vacuum impregnation method with surface modification is currently the main method for preparing CF thermal conductive composites. This method is simple and widely used, which can be used for large-scale production of CF composites. However, improving the impregnation efficiency of CF materials is the key to enhancing the thermal conductivity [63].

### 3.5. Suction Filtration Method

Suction filtration/vacuum filtration is a significant method for production of thermally conductive composites. Free-standing carbon films were thus developed. The adopted process contained the following steps: the Ag nanoparticles, few walled carbon nanotubes (FWCNTs) and mesophase pitch-based carbon fibers (MPCFs) were thoroughly stirred and mixed in presence of ethanol then treated by sonication for different times. Later, they were passed through a vacuum filtration and drying process, and as a result carbon-based films were produced. The schematic process is shown in Figure 7 [66].

The vacuum filtration is operated with the assistance of a vacuum pump. This method is simple to carry out and less demanding on equipment. However, this method is mainly used for the long CF composites, as a porous CF will reduce the vacuum during the filtration process and affect the impregnation efficiency. SiC_f_-CNTs/SiC composites were prepared by using carbon nanotube content of (25–30) vol% by a vacuum filtration method and it was revealed that this method solved the problems related to accumulation and introduction of low content CNTs in SiC/SiC composites. The formed composites were analyzed and passed through different characterizations which proved their enhanced thermal conductivity (23.9 W/m·K) at room temperature. The calculated thermal conductivity enhancement was 2.9 times, as compared to conventional SiC/SiC composites [67].

### 3.6. Electrophoretic Deposition Method (EPD)

The electrophoretic deposition (EPD) method is considered an effective method for material coating and improving the interfacial characteristics of composites. Compared with traditional spraying methods, such as dip coating and spin coating, the EPD method can quickly deposit reinforced fillers and effectively control the thickness of the reinforcing material on the substrate surface [68,69]. In recent years, the EPD method has been widely used in the field of CF composites [70]. Graphene oxide (GO) was deposited onto unmodified carbon fiber (CF) to produce GO/CFs by using an EPD method, such that CF was used as anode and GO was taken as cathode. The EPD method was applied for 10 min and which resulted an increased surface roughness in prepared sample [71]. A schematic of the experimental procedure is shown in Figure 8.

Sun et al. [5] attached graphene oxide (GO) to the surface of CFs by an EPD method, and adjusted the content of graphene on the CF by controlling the deposition time. GO was tightly covered and evenly distributed on the surface of the CF, which was beneficial to composite with the matrix. Li et al. [1] deposited the CF filaments by an EPD method in a solution of vertically aligned carbon nanotubes (VACNTs) with a concentration of 0.01 g/mL at a DC voltage of 30 V. According to the corresponding SEM images, the carbon nanotubes were firmly adhered to the surface of the CF filaments and the VACNT array was tightly bound by the matrix. The EPD method are mostly used for flat materials. When this method is used in the production of CF composite materials, there are high requirements for the flatness and thickness of the CF surface. The smooth surface is more beneficial to enhance the uniform distribution of the reinforced material on the surface and to enhance the thermal conductivity of the CF composite material. Talic et al. [72] deposited the M_n1.5_Co_1.5_O_4_ material on the steel substrate by EPD method. Due to the different thickness of the concave surface, the sintered coated steel substrate was prone to crack. The EPD method can effectively control the content of reinforced material by adjusting the concentration of the solution and the deposition time [73]. There are many references in which EPD method is utilized for preparing thermal conductive composites [71,74,75].

## 4. Influencing Factors of Thermal Conductivity

CF thermal conductive composites are mainly composed of CF, polymer matrix, and reinforced fillers such as Al_2_O_3_, BN, and graphene. Some factors, which may include CF type, content, reinforced thermal fillers, surface modification, orientation and CF 3D pathways, have a significant impact on the thermal conductivity of the resulting CF composites.

### 4.1. CF Types

At present, the precursors of CF are mainly derived from polyacrylonitrile (PAN) and pitch [76,77]. Among them, the output of the PAN CF accounts for about 85% of the total CF and its carbonization efficiency is about 50~60%. However, the pitch-based CF has the longer diameter and higher thermal conductivity, which is beneficial to the heat transfer [78,79]. Inoue et al. [80] used PAN CF (T300) and pitch-based CF (K223HG) to reinforce a ZrB2-SiC-ZrC (ZSC) matrix. The experiments indicated that the thermal conductivity of the composites prepared by the pitch-based CF was higher than that of PAN CF composites. However, the density of the PAN CF was lower and had better mechanical properties, and the existing preparation process of PAN CF is relatively advanced, which leads to the market dominance for the PAN CF [81]. The preparation process for PAN CF composite is shown in Figure 9.

The length of the CF will affect the thermal conductivity of the CF in the polymer matrix [83]. When the CF content is lower relatively, a longer CF is advantageous to form a more complete thermal chain, which promotes the heat transfer inside the materials [84]. However, when the CF content is higher relatively, the shorter CF is easily dispersed and forms the guiding structure [85]. Ghosh et al. [86] added the CF as the reinforcing material into a phenolic resin to make a composite bipolar plate. The influence of the CF length (1, 2, 3, 4 and 5) mm on the thermal conductivity of the composites was studied. The experiments indicated that when the CF content is 5 wt%, the thermal conductivity of the bipolar plate increased first and then decreased as the length increased and the thermal conductivity of the bipolar plate composite material is the highest when the CF length is 2 mm. Therefore, enhancing the thermal conductivity of the CF composites cannot rely entirely on increasing the CF length, and too long a CF will cause structural defects inside the composites. Fang et al. [87] reinforced the CF fabric with CF with lengths of 0.1 and 0.5 mm based on the DGEBA resin. From the SEM images, the shorter CFs formed a denser and oriented structure. The CFs with the length of 0.1 mm could easily exist in the gaps of the CF fabric, which promoted the thermal conductivity. When the CF content was 7.5 wt%, the thermal conductivity of the composites prepared from 0.1 and 0.5 mm CFs were 1.75 and 1.26 W/m·K, respectively. Therefore, the appropriate CF length should be selected according to the CF content and specific structure inside the composite material. A schematic process for preparation of pitch-based CF composite is shown in Figure 10.

### 4.2. CF Content

In order to avoid damage to the mechanical properties of the CF composite materials, the volume fraction of CF added into the polymer matrix is usually between 0 and 14 vol%. According to the data of Ma et al. [7] and Hou et al. [8], the thermal conductivity of CF composites increased with the increase of the volume fraction of CF, and the improvement of the CF composite material with a larger volume fraction is more obvious by the guide structure.

Cho et al. prepared CF composite materials with polyketone (PK) and chopped CFs with a mass fraction of 0~30 wt% by extrusion molding. Their experiments indicated that with the increase of the CF content, the thermal conductivity and Young’s modulus of the composite gradually increased. When the content of CF was 30 wt%, the thermal conductivity and Young’s modulus of the CF composite increased by 300% and 520%. Therefore, the increase of the CF content not only promotes the thermal conductivity, but also strengthens the mechanical properties of the material [89]. The effects of carbon content on thermal conductivity values are shown graphically in Figure 11.

### 4.3. Synergistic Effects

The reinforced thermal fillers for CF composites are usually oxides (Al_2_O_3_, MgO), nitrides (Si_3_N_4_, BN, AlN) and carbon materials (CNT, graphene) [92,93]. Although the content of the reinforced filler is restricted due to the large space occupied by the CF, the thermal conductivity of the fillers itself has a great influence on the CF composite. From the data of Wang [32] and Guo, it could be obtained that adding appropriate amount of reinforced thermal fillers on the basis of CF polymer composite significantly improved the thermal conductivity. Guo et al. added Ti_3_AlC_2_ and CF materials with the total mass of 30.2 wt% to the epoxy resin. The thermal conductivity of the composites increased first and then with the increase of the CF content it decreased. When the carbon fiber content was about 24.2 wt%, the thermal conductivity of the composite reached the maximum (9.68 W/m·K).

Zhao et al. [94] composited CF coated with copper and trimethylsilane with silicone rubber. When the content of copper particle-reinforced CF material was only 4 wt%, the thermal conductivity of the composite material reached 1.99 W/m·K, which is 2.1 times more than that of CF composite material without copper particles. This experiment showed that by adding a small amount of reinforced fillers, the thermal conductivity of the CF composite material was significantly improved under the condition that the mechanical properties were not damaged. Shi et al. [95] composited graphite, CF (TG800) and cyanate resin (CE). When the mass fraction of graphite was 0–10 wt%, the thermal conductivity of the composite with graphite and resin matrix increased from (0.26–0.46) W/m·K, and the thermal conductivity of the composite with CF, graphite and resin matrix increased from (0.6–1.36) W/m·K, an increase of 127%. The experiment indicated that as the graphite content increased, the thermal conductivity of CF composites also increased. Xu et al. electroplated copper particles onto the surface of acidified CF filaments. When the CF content was 29.34 vol%, the thermal conductivity reached 30.69 W/m·K. As the plating time increased, the Cu coated on the carbon fiber reduces the porosity of the material and affects the impregnation of the resin matrix, which will destroy the mechanical properties of the CF composites. The addition of excess Cu is also not conducive to the improvement of the thermal conductivity. The addition of excessive fillers will cause internal fractures in the composite material. Therefore, when preparing a CF thermal conductive composite, it is necessary to adjust the content of the reinforced fillers to meet the performance requirements of different equipment, and we need to improve the thermal conductivity of CF composites while keeping other properties from being damaged.

### 4.4. Surface Modification of CF

The interface binding performance between the CF and polymer materials has a great effect on the thermal conductivity of the CF composites. Jiao et al. [96] used an epoxy sizing agent to graft acrylamide onto the surface of CF and compound with vinyl ester resin (VE). The interfacial shear strength and interlayer shear strength of the CF composites prepared by surface modification increased by 86.96% and 56.61%. The optimization of the interfacial binding of the CF composites provides favorable conditions for the energy conduction. Wang et al. [97] configured a mixed solution of concentrated nitric acid (HNO_3_) and concentrated sulfuric acid (H_2_SO_4_) with a volume ratio of 3:1 and put the CF in the solution for acidification to compound with the resin. The results showed that the shear strength of the CF and epoxy resin surface increased with the extension of the acidification time, which indicated that the interfacial binding was improved and provided some help for reducing the interface thermal resistance of the composite materials. The shearing force generated between the screws of the conventional preparation of the chopped CF will cause damage to the CF. The protective treatment of the surface of the CF during the shearing process is beneficial to the performance of the CF itself and the composite. Hwang et al. [98] coated long fibers with a layer of thermoplastic resin by the dipping method and then cut them. Because the thermoplastic resin cures fast and has a certain fracture toughness [99], the damage generated during the shearing process will be reduced. The experiment indicated that the mechanical properties and thermal stability of the CF composites was better than that of the CF composites without surface modification [100].

Electrophoretic deposition (EPD), as a highly efficient surface treatment method that plays an important role in the preparation of CF thermal conductive composites. Zheng et al. [101] used Cu(NO_3_)_2_ as the electrolyte and CF fabric and copper plate as electrodes. BN powder was added to the electrolyte until the concentration of the BN was up to 5 g/L. Cu/BN reinforced CF fabric prepared by EPD method was annealed at 500 °C [102]. SEM pictures showed that Cu/BN was densely and uniformly distributed on the surface of the CF fabric to form a complete internal thermal conductive network. The thermal conductivity of the surface-treated CF composite in the plane direction was 6.14 W/m·K i.e., 47% higher than that of the composite without surface treatment, and increased by 217% (2.16 W/m·K) in the vertical direction.

Surface modification can be divided into two types, i.e., physical and chemical modifications. In physical modification the roughness of carbon fiber is increased by exposing it to some physical change, which increases its surface/reacting area for sticking with matrices, while in chemical modification, the carbon fibers are chemically modified by combining with certain functional groups which enhance its binding activity towards matrices. However, the carbon fibers used for composites preparation are usually capable of both modifications. Surface treatment of carbon fiber may enhance its efficiency in different aspects e.g., the sticking ability of CF is enhanced due to its wet behavior towards matrices, the Van-der-walls forces can be established which can considerably effect on amalgamation of CF and matrices and mechanical changes can significantly promote the interconnecting of CF towards matrices. Recently, different techniques have been adopted for surface modification of CF and can be listed as oxidation, plasma treatment, gamma treatment, rare earth element treatment and electrochemical treatment [103]. As shown in Figure 12.

### 4.5. Orientation of CF

Orientation plays a fundamental role for constructing high performance materials used for thermal management in electronics. Variation of thermal conductivity values in carbon fiber composites in different direction is highly dependent on the thermal pathways through carbon fiber-based composites, which is alternatively attributed to the orientation/alignment of carbon fiber as a filler as shown in Figure 13.

By using an electrostatic flocking method, vertically aligned carbon fibers were synthesized and used as a filler in thermal interface materials. It was anticipated that the through-plane thermal conductivity of the composites would be improved. Uniaxial CF- based composites were prepared and it was disclosed that the fiber angle has a significant impact in the formation of an effective geometry for substantial thermal conductivity values as the direction of the carbon fiber in the heat direction shows a significant enhancement in thermal conductivity. Fiber angles greater than 45° are considered suitable [58,105]. Hybrid fillers provide more continuous and conducting networks to polymer matrices, which are greatly dependent on the orientation among fillers, and effective alignment of CFs through other fillers on dispersion in polymer matrices results in efficient thermally conductive composites. Epoxy composites were prepared by dispersion of CF and alumina-based hybrid fillers and it was observed that the interfacial thermal resistance is effectively reduced, which is attributed to effective alignment of the CF through alumina [34]. Another hybrid system was developed by electroplating of hexagonal boron nitride and copper onto the CF using electrophoretic deposition technique and carbon fiber reinforced polymer composites (CFRP) were formed having improved through-plane and in-plane thermal conductivity values [103]. CF/melt-spun PMMS composites were synthesized by using different concentrations of CFs i.e., 6.3, 9.2 and 12.0 wt% respectively. Orientations and morphology of aligned CF in the PMMA composites were explained as shown in Figure 14 [106].

### 4.6. D Structure of CF

Due to its higher strength, low coefficient of thermal expansion (CTE), high stiffness and continuity, carbon fiber is considered a reliable source in electronics. The fused deposition modeling technique was adopted to prepare 3D CF-PEEK composites, SEM results showed presence of porosity and satisfactory pathways formed through composites for thermal conductivity. Figure 15 shows schematic process for alignment of CF in three dimensional CF-PEEK composites [107].

Unfortunately, in-plane thermal conductivity values are higher than through-plane in some carbon fibers, which is a barrier for carbon fiber reinforced polymer (CFRP) composites to be used in many electronic applications. Developing 3D structures of CFs for creating thermal pathways supporting the increase of the through-plane thermal conductivity is an option but the manufacturing methods give rise to complexities which demand new conditions to be set for CF utilization in electronics applications [108,109]. However, carbon fiber reinforced polymer (CFRP) composites were modified to generate 3D thermal conductive pathways for obtaining high through-plane thermal conductivity. For this purpose a z-filler-containing copper filling and aluminum coating was developed and adjusted into CFRP and it was demonstrated that the through-plane thermal conductivity of CFRP composite is enhanced [110]. Dimensional arrangement of CFs directly affects their usage rate at an industrial level for thermal interface materials preparation, as one dimensional CFs increase the viscosity resulting in preparation process complexity and thus three dimensional (3D) CF skeletons with vertically aligned CFs were introduced by using a freeze drying method, and by introducing them in an epoxy matrix highly arranged CF/epoxy composites were obtained showing efficient in-plane thermal conductivity and improved mechanical properties [7]. Table 1 shows thermal conductivity results for CF with different content reported in different articles.

## 5. Conclusions

This literature review mainly focuses on the role of CFs in the preparation of high thermal conductivity CF/polymer composites, as heat dissipation is a prominent aspect influencing the performance and durability of multifunctional electronic devices. Thus, high thermal conductivity materials are urgently needed to improve the efficiency of electronic devices. Different studies on thermal conductivity improvement of CF/polymer composites were discussed in detail. The advantages of CF as filler were summarized and it was demonstrated that CFs are regarded as a popular thermal conductive filler for manufacturing thermally conductive polymer composites due to some peculiar properties which include the ability of CFs to form highly orientated thermal conductive paths for composites in the vertical direction which provides a continuous thermally conductive path, and the fact CFs show synergistic behavior with other conductive fillers. Researchers have proved the benefits of using CFs in composites by making modifications, which led to higher thermal conductivity values, CFs have the ability to form three dimensional 3D pathways which depend on the method adopted for their alignment, volume percentage i.e., the fraction of CF directly affects the thermal conductivity of CF/polymer composites, and a high loading of oriented CF enhances the thermal conductivity. Moreover, thermal conductivity mechanisms, composite preparation methods and the factors influencing the thermal conductivity were discussed in detail. The purpose of this literature review is to summarize key features of thermally conductive CF/polymer composites, which may help further research in this field. Thus, it is worthwhile for us to work on the CF stripping and dispersion technologies in the future to prepare the high thermal conductivity CF/polymer composites with other excellent and comprehensive performance features.

## Figures and Tables

**Figure 1 polymers-13-00169-f001:**
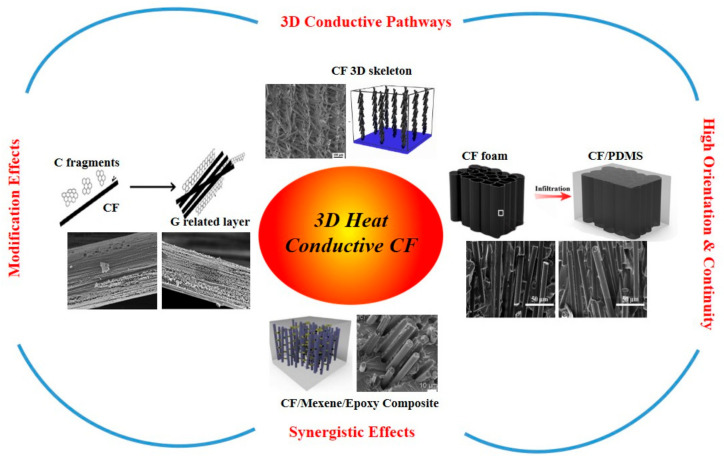
Overview of advantages of carbon fiber as a thermal conductive filler. Reproduced with permission from [7,8,9,10]. Copyright 2019 Elsevier B.V, 2020 Elsevier Ltd., The author(s) 2018, 2019 Elsevier B.V.

**Figure 2 polymers-13-00169-f002:**
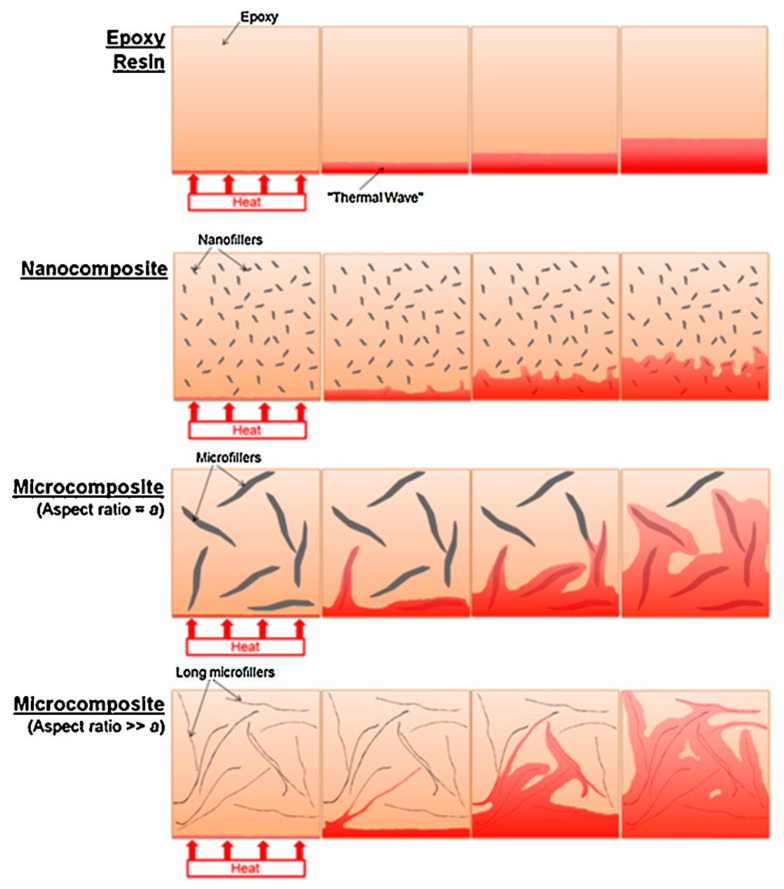
Schematic diagram illustrating the thermal conduction mechanism depending on size and aspect ratios in the composites. Reprinted with permission from [40]. Copyright 2016 Elsevier Ltd.

**Figure 3 polymers-13-00169-f003:**
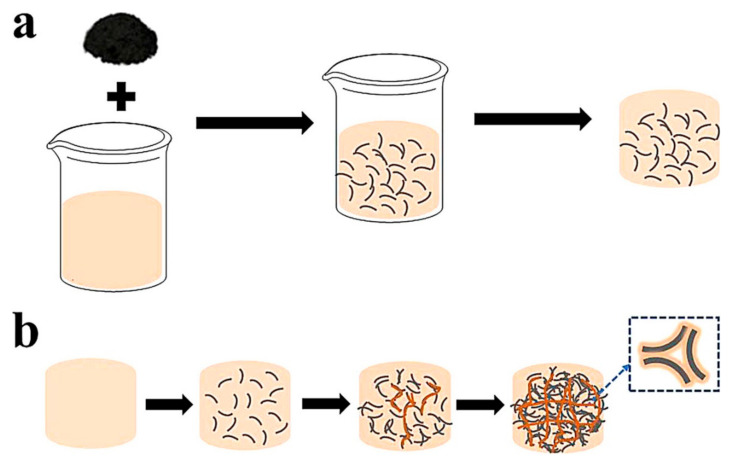
Schematic process of blending method for preparation of polymer composites. (**a**) The general concept for polymer composites preparation by using solution blending method. (**b**) Uniform dispersion of different filler contents in the polymer matrix. Reprinted with permission from [47]. Copyright 2020 Elsevier B.V.

**Figure 4 polymers-13-00169-f004:**
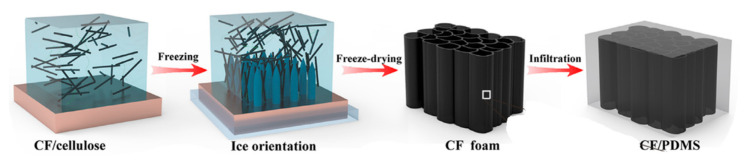
Schematic diagram for demonstrating the preparation process of the freeze-drying method for synthesizing orientated CF composite. Reprinted with permission from [8]. Copyright 2019 Elsevier B.V.

**Figure 5 polymers-13-00169-f005:**
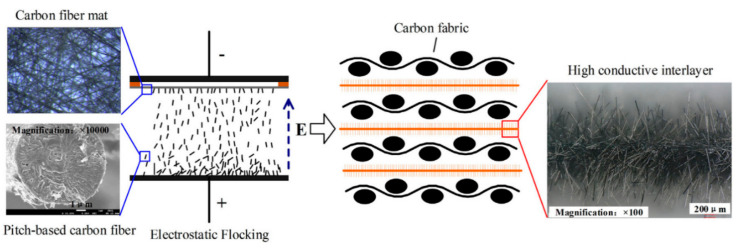
Schematic process of the electrostatic flocking method for CF mat preparation. Reprinted with permission from [57]. Copyright 2018 Elsevier Ltd.

**Figure 6 polymers-13-00169-f006:**
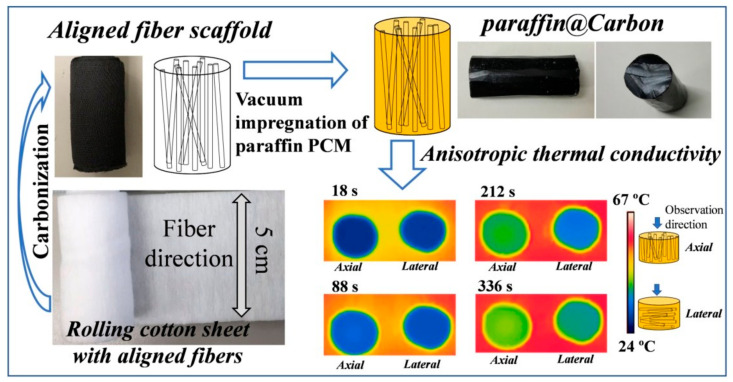
Schematic diagram for vacuum impregnation method to prepare carbon fiber based composites. Reprinted with permission from [63]. Copyright The Royal Society of Chemistry 2019.

**Figure 7 polymers-13-00169-f007:**
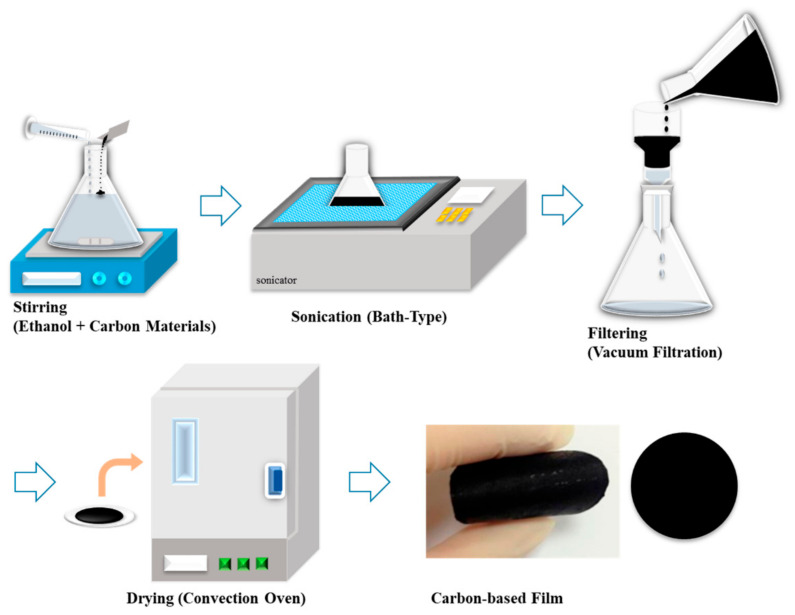
Schematic process of the suction filtration method for preparation of films based on carbon [66].

**Figure 8 polymers-13-00169-f008:**
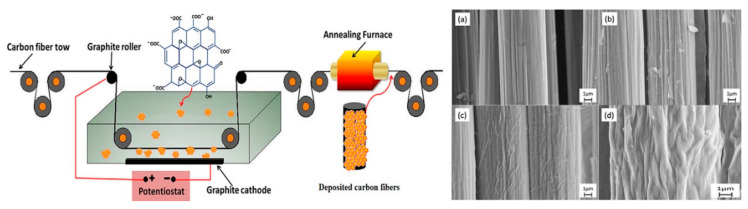
Schematic illustration of the electrophoretic deposition method for coating graphene oxide onto carbon fiber and (**a**–**d**) SEM images to show the results of EPD deposition at different voltages and current values. Reprinted with permission from [71]. Copyright 2012 Elsevier Ltd.

**Figure 9 polymers-13-00169-f009:**
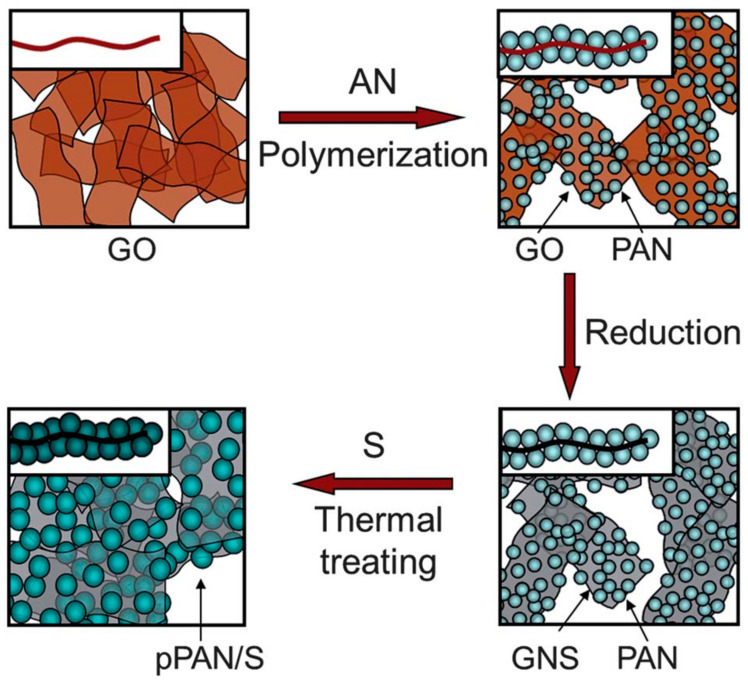
Preparation of polyacrylonitrile (PAN)-based CF composites. Reprinted with permission from [82]. Copyright The Royal Society of Chemistry 2012.

**Figure 10 polymers-13-00169-f010:**
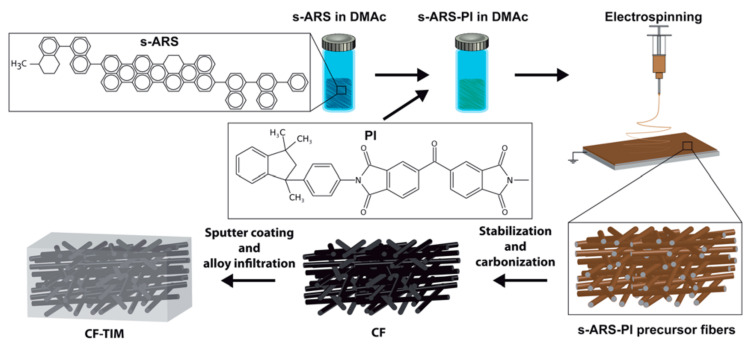
Synthesizing thermal conductive composite using pitch-based CF. Reprinted with permission from [88]. Copyright The Royal Society of Chemistry 2014.

**Figure 11 polymers-13-00169-f011:**
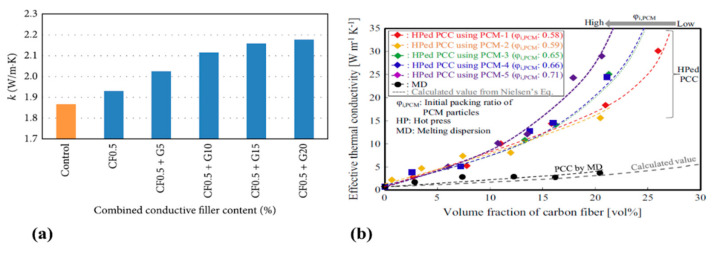
(**a**,**b**) shows the thermal conductivity enhancement by increasing CF content in different CF based composites. Reprinted with permission from [90,91]. Copyright 2015 Bai et al. and Elsevier Ltd.

**Figure 12 polymers-13-00169-f012:**
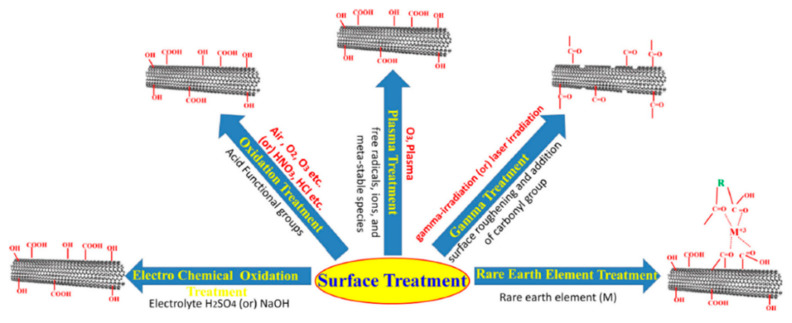
Diagram showing different techniques for CF surface modification. Reprinted with permission from [103]. Copyright 2018 Elsevier Ltd.

**Figure 13 polymers-13-00169-f013:**
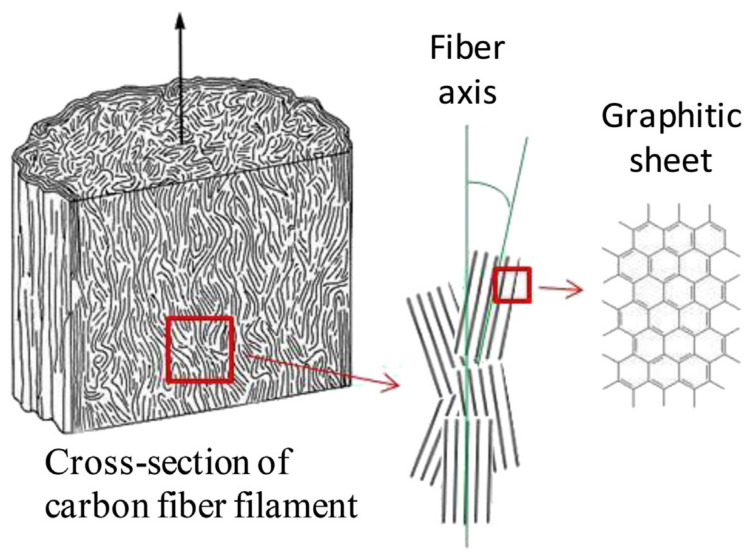
Shows orientation of carbon fiber in graphitic sheet for composite preparation. Reprinted with permission from [104]. Copyright 2016 Elsevier Ltd.

**Figure 14 polymers-13-00169-f014:**
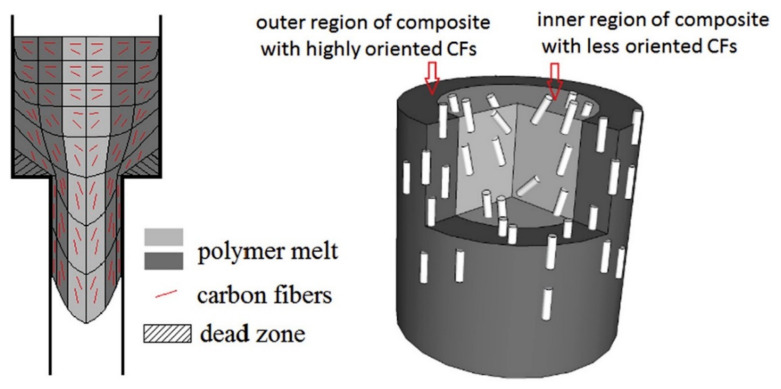
Schematic illustration of orientation and morphology of carbon fiber in different regions of CF/PMMS composites. Reprinted with permission from [106]. Copyright 2016 Elsevier Ltd.

**Figure 15 polymers-13-00169-f015:**
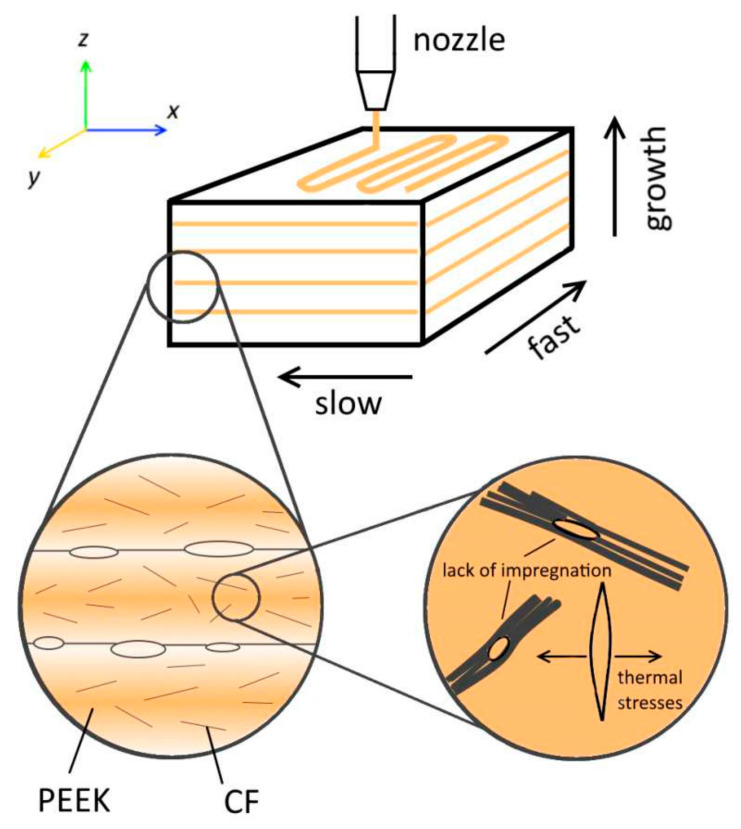
Schematic diagram for 3D printed CF-PEEK composites. Reprinted with permission from [107]. Copyright 2018 Elsevier Ltd.

**Table 1 polymers-13-00169-t001:** Thermal conductivities of Carbon fiber based composites at different fractions reported in different published references.

Filler	Fraction	TC (W/m·K)	Year	Reference
Chopped CF	13 wt%	2.84	2020	[7]
CF bundles	12.8 wt%	1.73	2020	[111]
CF + Al_2_O_3_	6.4 wt% CF + 74 wt% Al_2_O_3_	3.84	2020	[34]
Chopped CF	2 wt%	55	2019	[92]
Short CF	12.8 vol%	6.04	2019	[8]
CF felt + graphene oxide(GO)	4.25 wt%	0.53	2019	[48]
CF bundle + graphite sheet	60 vol% CF	6.20	2019	[31]
CF + flaked G	60 vol% CF + 10 wt% FG	0.6~1.36	2019	[95]
Chopped CF	18 wt%	2.95	2018	[112]
M-Cu-CF	4 wt%	1.99	2016	[94]

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
