# Peer review of "Preparation, Properties and Mechanisms of Carbon Fiber/Polymer Composites for Thermal Management Applications"

_polymers, 2021, doi:10.3390/polym13010169_

Round 1

Reviewer 1 Report

This review paper concerns a comprehensive overview of preparation methods of carbon fiber-polymer composites for thermal management applications. The paper is generally, well-structured, with a good number of figures, and a decent number of references. The manuscript may be of interest to the journal audience, but only after major changes. I have two issues; one major and one minor.

  • Minor: There are a few minor errors in the reference formatting (e.g.; issues with inconsistent capitalization, etc.).
  • Major: In my opinion, the language of the review is a strange combination of needlessly verbose and complex, coupled together with simplistic (and often valueless), sentences. This means that, while the material covered is fine, it is often  confusingly and/or unclearly conveyed such that the value to the reader is heavily reduced. 

As such, I recommend major changes of rewrites, before the manuscript can be accepted. 

Reviewer 2 Report

The Review work reported in this manuscript entitled (Preparation, Properties and Mechanisms of Carbon Fiber/Polymer Composites for Thermal Management Applications) is interesting and well presented. However, it needs improvements before acceptance. The work requires minor revision.

Comment 1: Authors need to add a list of abbreviations and a table of content in the revised manuscript.

Comment 2: There are some typographical errors in the manuscript, so authors need to correct them in the revised manuscript.

Comment 3: The authors need to add the graphical representation figure and explanation of publication trend (2000-2020) in the field of Carbon Fiber/Polymer Composites for Thermal Management.

Comment 4: In Figure 3, mentioned ‘a’ and ‘b’, but there is no description in the figure caption, so add the description of ‘a’ and ‘b’.

Comment 5: Figures 11 and 12 resolution is poor, so provide the figures with high-resolution.

Comment 6: Add the abbreviation of DGEBA, EGBDA in the manuscript text.
